# Analysis of the interval between submission and publication in genetics journals

**Rafael Leal Zimmer**[1,2]*, **Aline Castello Branco Mancuso**[3], **Ursula Matte**[4☯], **Patricia Ashton-Prolla**[1,4☯]

**1** Programa de Pós-Graduação em Medicina, Ciências Médicas, Universidade Federal do Rio Grande do Sul (UFRGS), Porto Alegre, Brazil, **2** Escritório de Projetos e Parcerias Estratégicas, Hospital de Clínicas de Porto Alegre (HCPA), Porto Alegre, RS, Brazil, **3** Unidade de Bioestatística–Diretoria de Pesquisa (DIPE), Hospital de Clínicas de Porto Alegre (HCPA), Porto Alegre, RS, Brazil, **4** Programa de Pós-Graduação em Genética e Biologia Molecular, Universidade Federal do Rio Grande do Sul (UFRGS), Porto Alegre, Brazil

☯ These authors contributed equally to this work.

* rzimmer@hcpa.edu.br

**Data Availability Statement:** All relevant data are available at: https://osf.io/3ep27/.

**Funding:** This study was partially funded by Fundo de Incentivo a Pesquisa – HCPA. No additional external funding was received for this study.

## Abstract

One of the main factors that attracts authors to choose a journal is the time interval between submission and publication, which varies between journals and subject matter. Here, we evaluated the time intervals between submission and publication according to journal impact factor and continent of author's affiliation, considering articles with authors from single or multiple continents. Altogether, 72 journals indexed in the Web of Science database within the subject matter "Genetics and Heredity", divided by impact factor into four quartiles and randomly selected were analyzed for time intervals from article submission to publication. Data from a total of 46,349 articles published from 2016 to 2020 were collected and analyzed considering the following time intervals: submission to acceptance (SA), acceptance to publication (AP) and submission to publication (SP). The median of the quartiles for the SP interval was 166 (IQR [118–225]) days for Q1, 147 (IQR [103–206]) days for Q2, 161 (IQR [116–226]) days for Q3 and 137 (IQR [69–264]) days for Q4, showing a significant difference among quartiles (p < 0.001). In Q4, median interval of time was shorter in interval SA but longer in interval AP, and overall, articles in Q4 had the shortest interval of time in SP. A potential association of the median time interval and authors' continent was analysed and no significant difference was observed between articles with authors from single versus multiple continents or between continents in articles with authors from only one continent. However, in journals from Q4, time from submission to publication was longer for articles with authors from North America and Europe than from other continents, although the difference was not significant. Finally, articles of authors from the African continent had the smallest representation in journals from Q1-Q3 and articles from Oceania were underrepresented in group Q4. The study provides a global analysis of the total time required for submission, acceptance and publication in journals in the field of genetics and heredity. Our results may contribute in the development of strategies to expedite the process of scientific publishing in the field, and to promote equity in knowledge production and dissemination for researchers from all continents.

**Competing interests:** The authors have declared that no competing interests exist.

# Introduction

In 2021, more than 3.6 million articles were published in journals indexed in the Web of Science database in various subject areas. Scientific articles, either online and/or on paper make new knowledge tangible and accessible and are published after a scientific peer review process that considers the impact of the topic on the field of study and the quality and novelty of the information that is presented [1]. The manuscript may be revised one or more times by the authors, or in case of rejection by the journal, be adapted for resubmission [2].

Agility in publishing scientific research results is important to optimize the dissemination of new knowledge and to stimulate the application of the scientific advances presented, according to van Teijlingen and Hundley [3]. These authors detailed the long road traveled for the publication of a methodological article, which in their example required submission to six journals until it was accepted for publication, and this process delayed publication for at least 1 year, as a result of a rigorous peer review.

Generally, the development of scientific research includes multiple stages until the publication of its results: hypothesis generation and study design, data collection and analysis, compilation and graphical representation of the results, manuscript writing, submission, and peer review. This complexity makes the process naturally time-consuming, and for this reason, authors often choose journals with greater agility in the evaluation process [4]. Between submission and publication of the article, delays are related to the editorial processes and subprocesses, including the time it takes to issue invitations to reviewers and to receive their reviews as well as the time needed by authors to respond to reviewers, revise the final version and resubmit their papers for publication [5–8]. One or more of these factors may contribute to an increase in time until publication.

## Delay contributing factors

The delay between submission and acceptance may be related to some problems present in the articles. These were highlighted by Chambers [9], such as: publication bias [10,11], insufficient statistical power [12], poor replicability [13], undisclosed analytic flexibility [14,15] and a lack of data transparency [16].

Another important aspect are factors related to the peer review process. According to Kriegeskorte [17], some main problems include: the review process is nontransparent [18], time-limited, and based on too few opinions; authors and reviewers operate under unhealthy incentives; evaluation delays publication; and the system is controlled by for-profit publishers and incurs in excessive costs. In addition to the problems found in the peer review process, the identification of niches in the post-publication review process were described, although these may also accelerate the dissemination of articles in the community [19].

## Publication time in life science

In the quest to make articles available to the community for quick presentation of results, we have seen an increase in the number of bibliometric studies related to areas of life sciences, focusing on evaluating the times of the publication process, seeking to identify the interval between submission, acceptance, and publication, as well as the total time between submission and publication (Table 1).

Increased scientific productivity has led to a greater number of submissions to scientific journals. Similarly, there are situations of great pressure for rapid publication, such as the scenario resulting from the COVID-19 pandemic. Finally, although increasingly used, the strategy of publications in the form of preprints is not yet universally accepted in the academic environment. All of these issues have increased the workload of journals, a scenario which can

**Table 1. Mean intervals between different stages of the publication process in life sciences.**

| Subject | Submission—Accepted | Accepted—Publication | Submission—Publication |
|---|---|---|---|
| Plastic Surgery [20] | 138 | 162 | 309 |
| Ophthalmology [21] | 133 | 100 | - |
| Family Medicine [22] | 177 | 192 | 369 |
| Clinical Trials [23] | 164 | 90 | 253 |

generate an increase in the time of review. However, the exacts reasons for the variability in review time observed among journals remain unknown [4]. This study aims to demonstrate the time from submission of a manuscript to its publication in journals within the same subject area and evaluates potential modulators of this time interval.

## Results and discussion

In total, 46,349 articles were collected from 72 journals in the field "Genetics and Heredity" listed in the *Web of Science* database between 2016 and 2020. The main characteristics of the data analyzed per JIF quartile are summarized in Table 2. We observed that the second quartile had the largest number of articles, and this may be related to the presence of journals with many published articles per issue in this quartile. As an example, one journal in the second quartile published 4,775 articles during the 5-year period considered in the present study. In addition, we observed that journals in the first quartile had a higher proportion of articles with authors from different continents (43%). This high rate of intercontinental cooperation may be related to the fact that articles published in high-impact journals are more complex and related to projects with higher execution costs, which could, in turn, require a larger group of authors or a consortium acting in partnership to optimize funding, human resources and also provide a multidisciplinary/multicentric interpretation of the results [24]. In the fourth quartile, journals with authors from multiple continents were less frequent.

Among the median values of the time intervals, articles in Q4 had the shortest SA time interval (63 days) and the longest AP interval (51 days). The distinct profile of articles in Q4 was also observed in histograms of the different time intervals (Fig 1), in which there was a concentration of SA intervals in the first 200 days in Q1-Q3, differing from Q4, which showed the highest concentration of SA intervals in the first 100 days. Journals in this quartile may be more agile in the article review and acceptance process; however, it is possible that they require

**Table 2. Main characteristics of the articles included and classified by JIF quartile.**

| JIF Quartile | Journals | Articles (%) | Articles by journal (interval) | IF[1] (interval) | CDC[2] (%) | Adj. Residual[3] (2 or more continents) |
|---|---|---|---|---|---|---|
| Q1 | 18 | 10,657 (23.0) | 95–2,562 | 4.878–11.361 | 43.6 | 32.5 |
| Q2 | 18 | 18,552 (40.0) | 142–4,775 | 3.412–4.797 | 29 | -6.9 |
| Q3 | 18 | 9,989 (21.6) | 157–1,850 | 2.166–3.260 | 32.5 | 4 |
| Q4 | 18 | ,7151 (15.4) | 63–2,369 | 0.226–1.891 | 14.2 | -33.1 |

[1] IF–JCR 2020.

[2] CDC—Articles with contributions from different continents.

[3] Adj. Residual–Adjust Residual representing the magnitude of association between 2 or more continents and JIF Quartile.

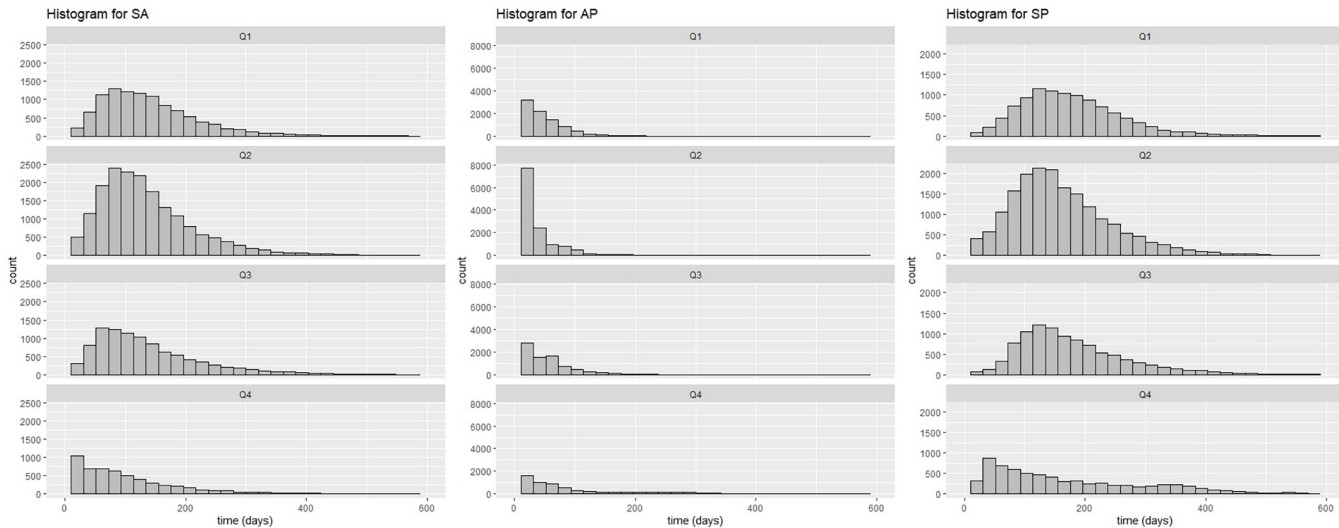

**Fig 1. Histogram of the time intervals between submission, acceptance, and publication by quartile[1].** [1]Note the difference in the Y-axis scale for AP interval.

a longer time to gather the articles necessary to constitute/complete an issue or volume of the journal, and for this reason, the AP interval is longer. In addition, according to Severin et al. [25], peer reviews in journals with a high impact factor tend to be more comprehensive. This finding agrees with a study by Gul et al. [26] showing a significantly positive relationship between the SA time intervals with the journal impact factor. Regarding the interval of time distributions, in days, the SP interval in Q4 did not have a normal distribution as observed in the other quartiles.

The numerical values of P25 and P75 between the intervals showed similar results between journals, which are constant in Q1, Q2, and Q3, and differ in Q4. Considering all intervals in the 4 quartiles there is a significant difference among groups (p < 0.001) (Table 3).

**Table 3. Time intervals (days) from submission to publication according to JIF quartile.**

| | JIF Quartile | | | | |
|---|---|---|---|---|---|
| | **Q1** | **Q2** | **Q3** | **Q4** | *p-value* |
| **Submission-Acceptance[1] (days)** | 125 | 118 | 115 | 63 | <0.001 |
| median | (1; 1,669; 81; 179) | (1; 1,046; 78; 172) | (1; 1,215; 72; 177) | (1; 1,854; 18; 124) | |
| (min; max; P25; P75) | | | | | |
| **Acceptance-Publicaton[2] (days)** | 32 | 18 | 34 | 51 | <0.001 |
| median | (1; 355; 14; 57) | (1; 753; 9; 35) | (1; 1,147; 12; 64) | (1; 774; 24; 106) | |
| (min; max; P25; P75) | | | | | |
| **Submission-Publicaton[3] (days)** | 166 | 147 | 161 | 137 | <0.001 |
| median | (2; 1,850; 118; 225) | (1; 1,048; 103; 206) | (2; 1,382; 116; 226) | (8; 2,227; 69; 265) | |
| (min; max; P25; P75) | | | | | |

[1] n = 45,214, distribution by quartile Q1–10,499; Q2–18,122; Q3–9,898; Q4–6,559.

[2] n = 45,196, distribution by quartile Q1–10,482; Q2–18,108; Q3–9,889; Q4–6,631.

[3] n = 45,657, distribution by quartile Q1–10,566; Q2–18,153; Q3–9,907; Q4–7,030.

* p-value significant by Kruskal-Wallis test.

Median time in the SP interval, considering all quartiles, ranged from 137 to 166 days. A study conducted by Siwek [22], based on a comparative analysis of data from journals in the field of family medicine, showed a time interval twice that obtained in our study. On the other hand, a study of journals in the field of plastic surgery showed SA, AP, and SP intervals of 138, 162, and 309 days, respectively which are longer than the time intervals observed here [20]. In another example of a study conducted with journals from the biomedical field, Andersen et al. [27] found comparable results to those presented here. Finally, in a similar study in the field of ophthalmology, the interval times of the first period, SA, had a median close to that observed in our study; however, in the second period, AP, the interval times observed here were 2 to 3 times shorter [21].

When we analyzed the potential impact of author origin (continents) on time intervals, per quartile, we did not observe relevant differences between times when comparing articles from North America and Europe and the other continents. However, for articles in Q4, the median time for the SP interval was longer in North America (133 days) and Europe (215 days) when compared to other continents (110 days), and the same occurred for the SA interval but this difference was not relevant (Table 4). The reasons for such differences, specific to articles in Q4 remain elusive and further studies are needed to confirm and clarify this finding.

**Table 4. Time from submission to publication journals by continent\*.**

| JIF Quartile | Variables | North America | | | Europe | | | Others | | |
|---|---|---|---|---|---|---|---|---|---|---|
| | | SA[1] | AP[2] | SP[3] | SA[1] | AP[2] | SP[3] | SA[1] | AP[2] | SP[3] |
| Q1 | Minimum | 1 | 1 | 2 | 1 | 1 | 4 | 1 | 1 | 3 |
| | Maximum | 1,669 | 353 | 1,850 | 887 | 282 | 901 | 907 | 355 | 955 |
| | Median | 121 | 36 | 165 | 121 | 38 | 167 | 133 | 18 | 162 |
| | Percentile 25 | 80 | 19 | 120 | 80 | 20 | 122 | 84 | 6 | 107 |
| | Percentile 75 | 175 | 59 | 223 | 173 | 64 | 224 | 189 | 42 | 229 |
| | N | 5,622 | 5,621 | 5,655 | 4,805 | 4,798 | 4,822 | 2,245 | 2,239 | 2,272 |
| Q2 | Minimum | 1 | 1 | 1 | 1 | 1 | 1 | 1 | 1 | 1 |
| | Maximum | 1,046 | 463 | 1,048 | 851 | 753 | 879 | 831 | 295 | 833 |
| | Median | 116 | 19 | 145 | 124 | 20 | 152 | 117 | 17 | 145 |
| | Percentile 25 | 75 | 10 | 100 | 82 | 10 | 106 | 79 | 7 | 102 |
| | Percentile 75 | 168 | 35 | 205 | 178 | 36 | 211 | 170 | 34 | 204 |
| | N | 5,545 | 5,553 | 5,562 | 6,375 | 6,378 | 6,387 | 7,789 | 7,768 | 7,799 |
| Q3 | Minimum | 1 | 1 | 6 | 1 | 1 | 6 | 1 | 1 | 2 |
| | Maximum | 678 | 1,147 | 1,147 | 1,215 | 577 | 1,382 | 1,199 | 467 | 1,211 |
| | Median | 104 | 47 | 158 | 121 | 28 | 165 | 122 | 20 | 160 |
| | Percentile 25 | 65 | 18 | 116 | 76 | 12 | 119 | 81 | 10 | 114 |
| | Percentile 75 | 166 | 70 | 222 | 187 | 63 | 232,5 | 182 | 51 | 224 |
| | N | 4,331 | 4,330 | 4,334 | 3,737 | 3,732 | 3,740 | 2,903 | 2,901 | 2,908 |
| Q4 | Minimum | 1 | 1 | 21 | 1 | 1 | 14 | 1 | 1 | 8 |
| | Maximum | 598 | 338 | 1,303 | 1,840 | 461 | 2,007 | 1,854 | 774 | 2,227 |
| | Median | 81 | 40 | 133 | 76 | 82 | 215 | 51 | 46 | 110 |
| | Percentile 25 | 33 | 20 | 87 | 42 | 27 | 115 | 12 | 24 | 56 |
| | Percentile 75 | 130 | 83 | 221 | 125 | 219 | 337 | 122 | 85 | 225 |
| | N | 879 | 882 | 986 | 1,654 | 1,656 | 1,914 | 4,237 | 4,305 | 4,371 |

\* The total N for each quartile will be higher, since there are articles with authors affiliated to both North American and European continents, thus accounted for in each of the groups.

SA[1] - Submission to Acceptance; AP[2] - Acceptance to Publication; SP[3] - Submission to Publication.

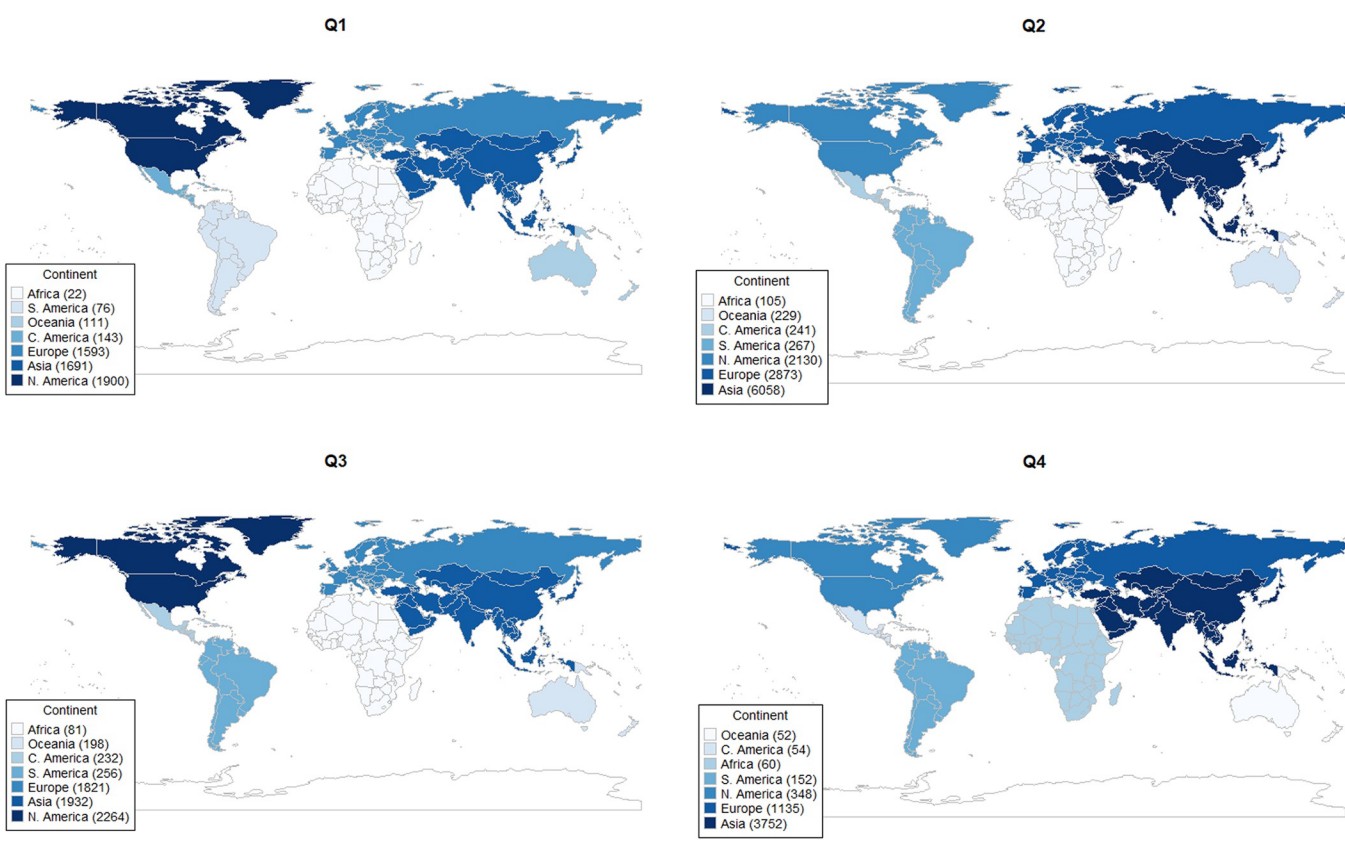

**Fig 2. Distribution of articles by continent (authors from a single continent).**

When analyzing articles with authors from a single continent, there was a higher number of articles from North America in Q1 and Q3, and a higher number of articles from the Asian continent in Q2 and Q4. In a similar study proposed by Dhoot [28], in the field of ophthalmology, the authors conducted a survey by country and showed the largest number of articles were from North America, followed by articles from the European and Asian continents. The continent with the lowest number of articles in Q1-3 was Africa and in Q4 was Oceania (Fig 2). Even though the present data prevents us from further exploring this apparent center-periphery phenomenon, others, such as Shelton [29], have connected the Lotka and Bradford laws of scientometrics to a proposed first and second laws of funding, able to forecast scientific production as a function of national funding of research and development (GERD). Our data may contribute to these analyses suggesting that the time frame for analysis may have to be taken into consideration, since the time between submission and publication varies according to the journal's impact factor and some countries have more papers in lower impact factor journals, at least in the field of genetics and heredity.

Several different factors may be explored as determinants of unequal research productivity ultimately resulting in heterogeneous quantitative and qualitative patterns of publications. These include, among others, the amount of resources invested in science production, the support a community receives to pursue research, the academic qualifications of researchers, institutional efforts to provide the infrastructure to promote research and allow protected time for research among the workforce, and also type or area of research may in itself influence scientific productivity. In addition, editorial policy may also be a significant determinant, especially in terms of the quality of peer-reviewed revisions and the time of manuscript evaluation.

Specific historical facts and scenarios may also have an impact on scientific production. As an example, the recent covid-19 pandemic has had a profound impact on publication dynamics, as reported by several authors from different regions in the world. In the last 2–3 years we have observed a significant and very rapid increase in the volume of covid-19 related publications, a faster mean time from submission to acceptance for COVID-19 papers and also a significant decrease in international collaborations. On the other hand, non-COVID-19 publications have suffered with this novel pattern, with a significant reduction of the number of non-COVID related publications and increase in publication times for these articles. In order to address these differences, with or without considering the COVID-19 pandemia, several efforts should be entailed including sustained financial resources for academic communities, especially in those geographic areas with less incentives, educational policies that promote the inclusion of research activities from early on in the education at different levels and engagement of local communities in the definition of research priorities and perhaps a more homogeneous approach in terms of editorial policy towards assessment of scientific production from different regions of the world [30–32].

As limitations of our study, we highlight that although we selected journals within the same subject area, this area is broad and relatively heterogeneous and contains subclassifications associated with animal genetics, human genetics, molecular genetics, plant genetics, genetic engineering, and evolution, among others. We were unable to compare the publication times between the journals of "Genetics and Heredity" and other biomedical areas, and there was no evaluation of the acceptance rate of the journals. There was no differentiation between the publication of the articles in their format (online or on paper) since the purpose was the temporal evaluation as to the speed until the results were made available to the public. Another limitation to be considered is related to the number of journals published per year, which was not considered for analysis. This was a cross-sectional study that covered a period of five years, with substantial differences in the number of articles recorded between the journals drawn for analysis.

## Conclusions

When comparing the median of the time intervals SA, AP and SP, we identified differences among journals from different JIF quartiles (p < 0.001), and although many factors may contribute to this difference, complexity of the study, funding, cooperative effort, multicenter versus single center initiatives may all be involved in a possible explanation to this observation. We also evaluated potential differences in processing times of articles according to authors' affiliations. We first compared articles with authors from single versus multiple continents and then we analyzed, among articles with authors from a single continent, differences between continents. Overall, we did not find a significant difference in our analyses. We can also highlight that for the articles with authors from a single continent, the African continent, although present in quartiles 1–3, had the lowest number of articles when compared to authors from other continents. Articles from the North American continent were well represented in the first and third quartiles, and articles from the Asian continent were well represented in the second and fourth quartiles. Among the main findings, we highlight that after an exhaustive review of more than 45,000 articles in the area "Genetics and Heredity", significant differences were identified in the SA, AP, and SP time intervals between journals from different JIF quartiles. Our results may contribute to the development of strategies to expedite the process of scientific publishing in the field of Genetics and Heredity and to promote equity in knowledge production and dissemination for researchers from all continents.

Understanding the scenario and identifying the bottlenecks of publishing research results through descriptive analyses as we have presented here, is the first step to propose change, and

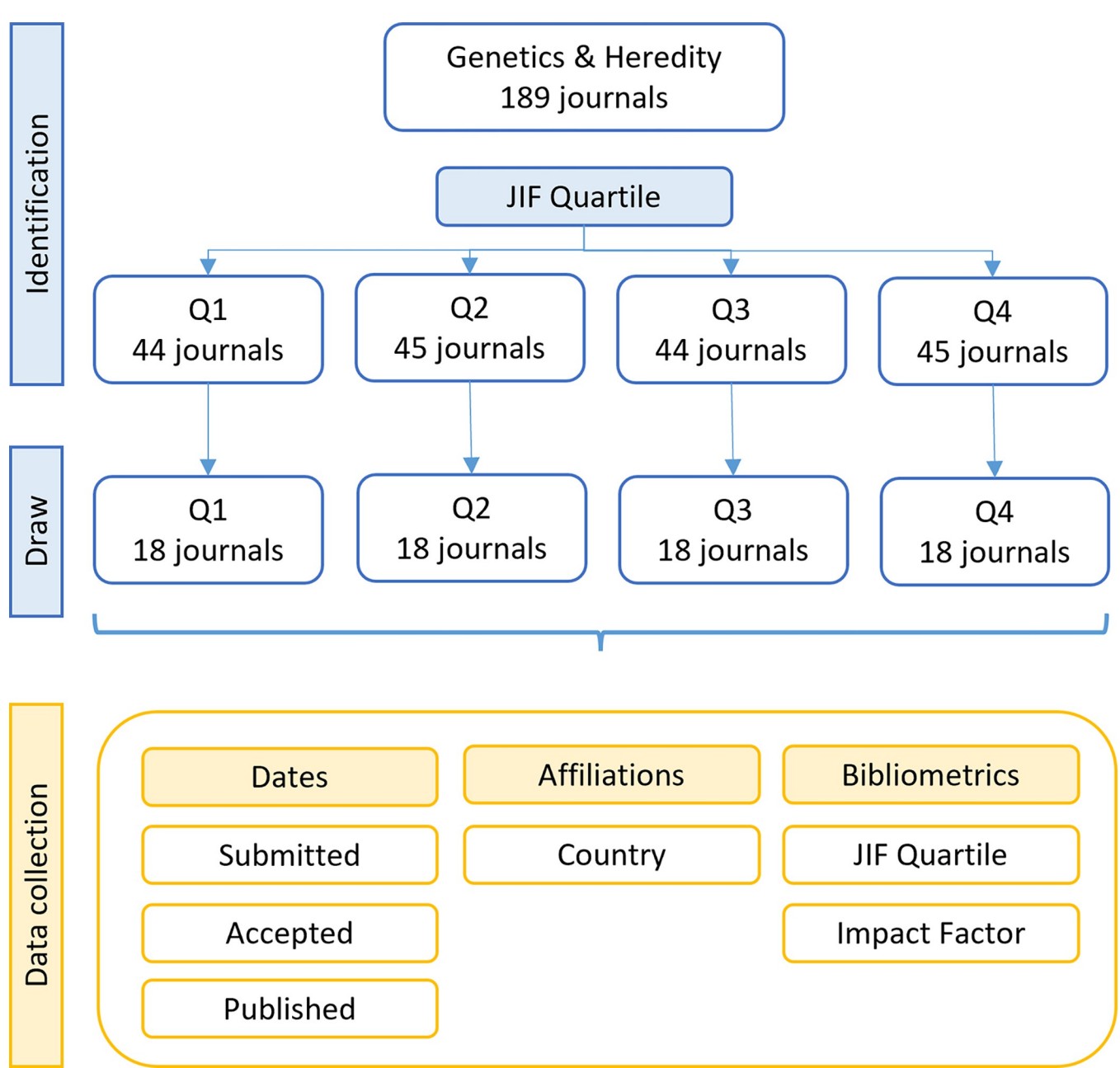

**Fig 3. Schematic representation of study design and data/results obtained in the different phases of the study.**

to develop novel strategies. We expect that this initial exploratory analysis will provide the basis for further research in the field and instrument editorial officers, policy makers and science/technology officials in the establishment of strategic actions to promote scientific publications in the field and originaing in different regions of the world.

## Materials and methods

The sample size was calculated using the online version of Statistical Package for the Social Sciences (SPSS). To estimate the average processing time, a relative margin of error between the

estimate and the unknown value of the population of 10% and a confidence level of 95% were considered. Considering a simple stratified sampling process in Q1, Q2, Q3, and Q4, with population sizes of 44, 45, 44, and 45 individuals, expected means of 281.5, 274.8, 239.2, and 449.4 days and expected standard deviations of 171.4, 147.8, 152.2 and 154 days [5], respectively, reached a total sample size of 64 journals divided into quartiles as follows: 16 in the Q1 stratum, 16 in the Q2 stratum, 16 in the Q3 stratum and 16 in the Q4 stratum. Adding 10% for possible losses and incomplete data, the estimated sample size was 18 for each stratum.

Journals with an impact factor (JIF) were selected in the subject area "Genetics and Heredity" in the Journal Citation Reports (JCR) 2020 database (Clarivate Analytics) and classified in quartiles according to their impact factor. Within each quartile, journals were randomly chosen up to a limit of 18. For each journal, the availability of information regarding the dates of submission (S), acceptance (A), and publication (P) was evaluated, and if it was not complete, the journal was removed from the analysis, and a new draw was performed until the minimum number of journals with complete information was reached (Fig 3). Based on the ISSN of the selected journals, the articles were collected from 2016 to 2020 in JCR. This study was approved by the Research Ethics Committee of Hospital de Clínicas de Porto Alegre.

Two strategies were used to collect information related to the journals: (a) using the web scrapping method in RStudio, i.e., with the use of codes to capture the specific information, or (b) retrieving the articles in PDF format manually in those cases where web scrapping was not possible due to lack of information on the internet. Variables analyzed were dates of submission, acceptance and publication, quartile of the journal in the JCR, impact factor, and affiliation (country) of the authors. Data on the author's country of origin were extracted using RStudio libraries and respective countries were grouped by continent. For the purposes of calculating the time intervals between quartiles, the following intervals were defined: submission to acceptance (SA), acceptance to publication (AP), and submission to publication (SP). If any of these dates were missing, the interval was calculated only for the available dates (e.g. SP only). Variations greater than 0 and smaller than 40,000 days were considered for the time interval analyses.

Data were compiled in Microsoft Excel® (Redmond, Washington), and the descriptive analyses were performed with SPSS® (version 23.0, Chicago, Illinois). Descriptive statistics were used to report the data, including median and interquartile range (IQR) for continuous variables and percentages for categorical variables. The time variables were represented by median and percentiles, and the comparison data between the groups were analyzed using the Kruskal–Wallis test between the quartiles and their time intervals, rejecting the null hypothesis ($p < 0.01$).

## Author Contributions

**Conceptualization:** Rafael Leal Zimmer, Ursula Matte, Patricia Ashton-Prolla.

**Data curation:** Rafael Leal Zimmer, Ursula Matte, Patricia Ashton-Prolla.

**Formal analysis:** Rafael Leal Zimmer, Aline Castello Branco Mancuso, Ursula Matte, Patricia Ashton-Prolla.

**Investigation:** Rafael Leal Zimmer, Ursula Matte, Patricia Ashton-Prolla.

**Methodology:** Rafael Leal Zimmer, Aline Castello Branco Mancuso, Ursula Matte, Patricia Ashton-Prolla.

**Project administration:** Rafael Leal Zimmer, Ursula Matte, Patricia Ashton-Prolla.

**Resources:** Rafael Leal Zimmer, Ursula Matte, Patricia Ashton-Prolla.

**Software:** Rafael Leal Zimmer, Aline Castello Branco Mancuso, Ursula Matte, Patricia Ashton-Prolla.

**Validation:** Rafael Leal Zimmer, Aline Castello Branco Mancuso, Ursula Matte, Patricia Ashton-Prolla.

**Visualization:** Rafael Leal Zimmer, Aline Castello Branco Mancuso, Ursula Matte, Patricia Ashton-Prolla.

**Writing – original draft:** Rafael Leal Zimmer, Aline Castello Branco Mancuso, Ursula Matte, Patricia Ashton-Prolla.

**Writing – review & editing:** Rafael Leal Zimmer, Aline Castello Branco Mancuso, Ursula Matte, Patricia Ashton-Prolla.

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
