## [Decision Letter · Decision Letter 0]

20 Dec 2022

PONE-D-22-25394Analysis of the interval between submission and publication in genetics journalsPLOS ONE

Dear Dr. Zimmer,

Thank you for submitting your manuscript to PLOS ONE. After careful consideration, we feel that it has merit but does not fully meet PLOS ONE’s publication criteria as it currently stands. Therefore, we invite you to submit a revised version of the manuscript that addresses the points raised during the review process.

We look forward to receiving your revised manuscript.

Kind regards,

Alejandro Vega-Muñoz, Ph.D.

Academic Editor

PLOS ONE

Journal Requirements:

"No"

4. We note that you have stated that you will provide repository information for your data at acceptance. Should your manuscript be accepted for publication, we will hold it until you provide the relevant accession numbers or DOIs necessary to access your data. If you wish to make changes to your Data Availability statement, please describe these changes in your cover letter and we will update your Data Availability statement to reflect the information you provide

Reviewers' comments:

Reviewer's Responses to Questions

**Comments to the Author**

1. Is the manuscript technically sound, and do the data support the conclusions?

Reviewer #1: Partly

Reviewer #2: No

Reviewer #3: Yes

2. Has the statistical analysis been performed appropriately and rigorously? 

Reviewer #1: Yes

Reviewer #2: No

Reviewer #3: Yes

3. Have the authors made all data underlying the findings in their manuscript fully available?

Reviewer #1: Yes

Reviewer #2: No

Reviewer #3: Yes

4. Is the manuscript presented in an intelligible fashion and written in standard English?

Reviewer #1: Yes

Reviewer #2: Yes

Reviewer #3: Yes

5. Review Comments to the Author

Reviewer #1: Dear authors,

Thanks for letting me read and review your article.

Although your article has many exciting features, it demands the publication of some necessary corrections.

First, and referring to the literature review and problem-oriented introduction, there needs to be more problematization about productivity, availability of reviewers, and other connected factors that should be presented in this section of your paper. For example, some work had been reflected in the productivity of the peer review process. Articles like "Teixeira da Silva, J. A., & Dobránszki, J. (2015). Problems with traditional science publishing and finding a wider niche for post-publication peer review. Accountability in research, 22(1), 22-40." could serve to produce a more detailed and theory-driven literature review for your analysis and later discussion.

Later on, when the paper goes to the results and discussion, the article uses several studies to compare the timing from submission to publication. These studies deserve their own space in a literature or problem-oriented literature section. To use such work in your results and discussion appears not grounded in the line of argument of your article.

Finally, when the paper goes into the crucial discussion about the article by continents, authors refrain from getting involved with essential discussions in knowledge production that happened in the last years—for example, center-periphery in knowledge production and global knowledge circulation phenomena. If the paper wants to discuss such topics, it is necessary to review the literature about it and confront it in this section of the article.

Finally, you mention, "Our results may contribute in the development of strategies to expedite the process of scientific publishing in the field, and to promote equity in knowledge production and dissemination for researchers from all continents." Developing strategies to promote equity in knowledge production is an important issue that connects with your results that you could have expanded in a better sense. Therefore, such strategy developments deserve at least a few paragraphs in your conclusions and policy recommendations.

All in all, your paper presents a critical analysis that demands more engagement with the literature and, consequently, to improve your article's discussion and conclusions.

Reviewer #2: The present research intends to show an important part of scientific productivity such as publication time. In order for the work not to be considered as a simple and not very detailed description, it should have an adequate statistical rigor, showing the work of cleaning the databases, what criteria are considered for the elimination of missing or lost data, protocols of the tradition and current paradigms of scientometrics and an exhaustive review of the literature to consider which are the statistical methodologies most used for this purpose. In that sense, this article precisely misses these points. Despite its weakness, it is recommended to enhance its analysis with more advanced statistics for its analysis. There are countless laws from scientometrics that could help to pose more challenging questions with deeper and richer analysis. The analysis of quartiles is interesting but only as a descriptor factor; however, the real problem is in the limited challenge of the research

Reviewer #3: This is a well written and innovative paper. The authors developed a very original idea and use a interesting methodology to study it. The conclusions are relevant and this paper may be very sueful to PlosOne readers.

6. PLOS authors have the option to publish the peer review history of their article (what does this mean?). If published, this will include your full peer review and any attached files.

Reviewer #1: **Yes: **Juan Felipe Espinosa-Cristia

Reviewer #2: No

Reviewer #3: No

---

## [Author Response · Author response to Decision Letter 0]

18 Feb 2023

Reviewer #1: 

Dear authors,

Thanks for letting me read and review your article.

Although your article has many exciting features, it demands the publication of some necessary corrections.

First, and referring to the literature review and problem-oriented introduction, there needs to be more problematization about productivity, availability of reviewers, and other connected factors that should be presented in this section of your paper. For example, some work had been reflected in the productivity of the peer review process. Articles like "Teixeira da Silva, J. A., & Dobránszki, J. (2015). Problems with traditional science publishing and finding a wider niche for post-publication peer review. Accountability in research, 22(1), 22-40." could serve to produce a more detailed and theory-driven literature review for your analysis and later discussion.

Response: Thank you very much for this comment. We have adjusted the introduction and discussion according to your suggestions.

Later on, when the paper goes to the results and discussion, the article uses several studies to compare the timing from submission to publication. These studies deserve their own space in a literature or problem-oriented literature section. To use such work in your results and discussion appears not grounded in the line of argument of your article.

Response: Thank you very much for this comment. We have removed the studies/references from the results section and adjusted the discussion as mentioned previously.

Finally, when the paper goes into the crucial discussion about the article by continents, authors refrain from getting involved with essential discussions in knowledge production that happened in the last years—for example, center-periphery in knowledge production and global knowledge circulation phenomena. If the paper wants to discuss such topics, it is necessary to review the literature about it and confront it in this section of the article.

Response: Thank you for your suggestion, however we feel that this is beyond the scope of the present article. The main objective is to first describe the distribution of production in the continents for each quartile and bring attention to the problem. On the other hand, a more profound study of the determinants of such results is certainly warranted and should include the aspects mentioned in your comment, such as the recently observed center-periphery phenomenon as well as other current scenarios such as, for instance, evident differences between covid vs. non-covid related publications . We have mentioned this aspect among the limitations of the study. 

Finally, you mention, "Our results may contribute in the development of strategies to expedite the process of scientific publishing in the field, and to promote equity in knowledge production and dissemination for researchers from all continents." Developing strategies to promote equity in knowledge production is an important issue that connects with your results that you could have expanded in a better sense. Therefore, such strategy developments deserve at least a few paragraphs in your conclusions and policy recommendations. All in all, your paper presents a critical analysis that demands more engagement with the literature and, consequently, to improve your article's discussion and conclusions.

Response: Thank you very much for this comment. We have adjusted the conclusions according to your suggestions.

Reviewer #2: The present research intends to show an important part of scientific productivity such as publication time. In order for the work not to be considered as a simple and not very detailed description, it should have an adequate statistical rigor, showing the work of cleaning the databases, what criteria are considered for the elimination of missing or lost data, protocols of the tradition and current paradigms of scientometrics and an exhaustive review of the literature to consider which are the statistical methodologies most used for this purpose. In that sense, this article precisely misses these points. Despite its weakness, it is recommended to enhance its analysis with more advanced statistics for its analysis. There are countless laws from scientometrics that could help to pose more challenging questions with deeper and richer analysis. The analysis of quartiles is interesting but only as a descriptor factor; however, the real problem is in the limited challenge of the research

Response: Thank you very much for your comments. Regarding cleaning of databases, it was performed in two steps. First, journals that do not provide submission (S), acceptance (A), and publication (P) dates were excluded from the sample, with a new one drawn from the original pool. For manuscripts not presenting all dates, the interval was calculated only for the available dates (e.g. SP only). Therefore, we have slightly different numbers in each interval, as presented in the footnote of table 3. We added a comment on the study proposed by Shelton (2020) relating Lotka's Law and Bradford's Law to national funding, and suggested that the time frame chosen for analysis (e.g. 1 year after funding) may bias the results towards higher impact journals, in which some countries are under-represented. 

Reviewer #3: This is a well written and innovative paper. The authors developed a very original idea and use a interesting methodology to study it. The conclusions are relevant and this paper may be very useful to PlosOne readers.

Response: Thank you very much for your comments.

---

## [Decision Letter · Decision Letter 1]

11 Apr 2023

Analysis of the interval between submission and publication in genetics journals

PONE-D-22-25394R1

Dear Dr. Zimmer,

We’re pleased to inform you that your manuscript has been judged scientifically suitable for publication and will be formally accepted for publication once it meets all outstanding technical requirements.

Kind regards,

Alejandro Vega-Muñoz, Ph.D.

Academic Editor

PLOS ONE

Additional Editor Comments (optional):

Reviewers' comments:

Reviewer's Responses to Questions

**Comments to the Author**

1. If the authors have adequately addressed your comments raised in a previous round of review and you feel that this manuscript is now acceptable for publication, you may indicate that here to bypass the “Comments to the Author” section, enter your conflict of interest statement in the “Confidential to Editor” section, and submit your "Accept" recommendation.

Reviewer #1: All comments have been addressed

Reviewer #2: All comments have been addressed

2. Is the manuscript technically sound, and do the data support the conclusions?

Reviewer #1: Yes

Reviewer #2: Yes

3. Has the statistical analysis been performed appropriately and rigorously? 

Reviewer #1: Yes

Reviewer #2: Yes

4. Have the authors made all data underlying the findings in their manuscript fully available?

Reviewer #1: Yes

Reviewer #2: Yes

5. Is the manuscript presented in an intelligible fashion and written in standard English?

Reviewer #1: Yes

Reviewer #2: Yes

6. Review Comments to the Author

Reviewer #1: Thanks again for let me read your reviewed version of the article. I do think that you had covered the full comments and suggestions that I made before.

Reviewer #2: (No Response)

7. PLOS authors have the option to publish the peer review history of their article (what does this mean?). If published, this will include your full peer review and any attached files.

Reviewer #1: **Yes: **Juan Felipe Espinosa Cristia, PhD.

Reviewer #2: **Yes: **Nicolás Contreras-Barraza

---

## [Editor Report · Acceptance letter]

26 Apr 2023

PONE-D-22-25394R1 

Analysis of the interval between submission and publication in genetics journals 

Dear Dr. Zimmer:

I'm pleased to inform you that your manuscript has been deemed suitable for publication in PLOS ONE. Congratulations! Your manuscript is now with our production department. 

Kind regards, 

on behalf of

Dr. Alejandro Vega-Muñoz 

Academic Editor

PLOS ONE